



# Impacts of wind field characteristics and non-steady deterministic wind events on time varying main-bearing loads

Edward Hart[1], Adam Stock[1], George Elderfield[1], Robin Elliott[2], James Brasseur[3], Jonathan Keller[4], Yi Guo[4], and Wooyong Song[5]

[1]Wind Energy and Control Centre, Department of Electronic and Electrical Engineering, The University of Strathclyde, Glasgow, UK
[2]Onyx Insight, Nottingham, UK
[3]University of Colorado, Boulder, United States
[4]National Renewable Energy Laboratory, United States
[5]ORE Catapult, Inovo, Glasgow, UK

**Correspondence:** Edward Hart (edward.hart@strath.ac.uk)

**Abstract.** This work considers the characteristics and drivers of the loads experienced by wind turbine main-bearings. Simplified load response models of two different hub and main-bearing configurations are presented, representative of both inverting direct-drive and four-point mounted geared drivetrains. The influences of deterministic wind field characteristics, such as wind speed, shear, yaw offset and veer, on the bearing load patterns are then investigated for similarity scaled 5, 7.5 and 10MW reference wind turbine models. Main-bearing load response in cases of deterministic gusts and extreme changes in wind direction are also considered for the 5MW model. Perhaps surprisingly, veer is identified as an important driver of main-bearing load fluctuations. Upscaling results indicate that similar behaviour holds as turbines become larger, but with mean loads and load fluctuation levels increasing at least cubically with the turbine rotor radius. Strong links between turbine control and main-bearing load response are also observed.

## 1 Introduction

Wind energy technology is seeing a fundamental shift towards larger (3-15 MW) offshore turbines located further from shore (Barter et al., 2020). Cost reductions necessary for ensuring the financial viability of future offshore wind projects are in turn driving a need for improved reliability and minimising repair requirements and associated costs of access and downtime. Direct-drive technology, it is argued, will see reduced maintenance costs as a result of there being fewer parts and thus higher reliability than a geared drivetrain, offsetting any increase in capital cost related to high direct-drive generator weight and reliance on a large amount of expensive, rare-earth magnets. In conjunction with these developments, advances in rotor support technology are taking place that support and transfer the majority if not all rotor loads other than torque to the bedplate, isolating the rest of the drivetrain from these effects. Geared drivetrains increasingly are supported by two main-bearings within an integrated housing for example by Vestas (Demtröder et al., 2019; Nejad et al., 2021), while a few geared and some direct-drives use an "inverting" configuration, in which the entire hub rotates about a stationary internal mounting with one more more main-bearings (Guo et al., 2014, 2015; van Dam, 2020; Torsvik et al., 2018; Hart et al., 2020; Nejad and Torsvik,





2021). Such technology is currently implemented in Siemens Gamesa and GE direct-drive turbines, among others. The main-bearings provide a rotationally free support between the main shaft and bedplate and are an operations critical component for these machines.

Higher-than-expected main-bearing failure rates in some turbine populations have led to increased research focus in recent years (Sethuraman et al., 2015; Keller et al., 2016; Hart et al., 2019, 2020). Findings from these studies indicate that operational load conditions for these components are complex and dynamic in nature (Hart, 2020; Guo et al., 2021), deviating from the conditions implicitly assumed by current design standards which evolved out of more conventional rolling bearing applications (Nejad et al., 2021). These issues are compounded by the inherently multidisciplinary nature of main-bearing function. Aside

from rotor and drivetrain weight, the aerodynamic loads on main-bearings are largely driven by the non-steady aerodynamic and control responses to the continual passage of strong turbulence motions in the wind field. Crucially, main-bearings are rotating tribological components designed to achieve the above while maintaining a separation of internal surfaces via a lubricant film at relatively low rotational speeds compared to many applications. A detailed scientific understanding of each of the links in the above chain, including their interactions, is therefore necessary to ensure main-bearing reliability. The current

research contributes to this effort by identifying characteristics and key drivers of main-bearing loading present in two different main-bearing support configurations. This is achieved using simplified main-bearing load response models in conjunction with aeroelastically derived hub loading from simulations in steady mean and non-steady wind conditions. We extend our analysis to larger wind turbines by upscaling under the assumption of similarity.

    Section 2 provides relevant background to this research, including a discussion of previous work on main-bearings in wind

turbines. The methodology is then detailed in Section 3, including model derivations and parameter selection. Results of the analyses are presented in Section 4, with findings then summarised and discussed in Section 5.

## 2   Background

In order to facilitate a detailed investigation of the various factors influencing main-bearing loads, relevant literature is reviewed and key aspects of the aeroelastic models used in the study discussed, including upscaling and control considerations.

### 2.1   Main-bearing modelling and load characteristics

Several previous efforts have focused on main-bearing load characteristics in 1.5-2 MW geared machines. In Hart et al. (2019), mean and peak main-bearing radial loads were extracted from aeroelastic simulations and analytical representations of geared three- and four-point support configurations of a 2 MW turbine. The analysis indicated that wind field characteristics are a strong driver of main-bearing mean and peak non-steady loading, with different drivetrain configurations found to differ in

load response. Ratios of axial-to-radial main-bearing loading were found to be higher for the three-point mounting, attributed to relatively larger radial loads occurring in four-point support cases (due to moments being reacted across a shorter torque-arm). Axial motion, roller loads, cage slip, and bearing outer ring strains have also been examined in an instrumented 1.5 MW three-point support geared turbine and compared to analytical and finite element models (Guo et al., 2021; Bergua et al., 2021).





Hart (2020) considered the time-varying nature of main-bearing radial loading on a similar turbine, identifying repeating (also
called 'looped') load patterns linked to the repeated passing of turbine blades through wind field structures (Lavely, 2017;
Nandi et al., 2017; Hart, 2020). In the same work, an internal load model of a double-row spherical roller main-bearing was
developed and used to assess impacts of identified load structures on individual rollers. The presence of identified radial load
loops, combined with thrust loading, was shown to drive significant variations in individual roller loads, with rollers seen
to experience load fluctuations of 40 and even $> 60$ kN in a matter of 1-2 seconds. These studies all considered only the
case of a non-moment reacting support at the main-bearing, a condition that generally holds for a double-row spherical roller
bearing (SRB) in a three-point mounting configuration but does not hold for paired tapered roller bearings (TRBs) which, as
a unit, provide a combination of force and moment reactions to applied loads. Stirling et al. (2021) proposed an extension of
the preceding analytical main-bearing representations that accounts for moment reactions via inclusion of rotational springs.
A method by which appropriate spring stiffnesses can be extracted from finite element representations of a given drivetrain
was also presented. Simplified 3D finite element models, each exhibiting reaction properties of double-row SRB or TRB
supports, were used to demonstrate the effectiveness of the outlined approach. Results of the study also indicated that, in
the non-moment-reacting SRB case, analytical models used to date provide accurate representations of loading at the main-
bearing when compared to the reaction loads calculated by the finite element model. It should be noted that some effects, such
as bedplate flexibility, were not included in the finite element models. Other important research has been conducted which
considers main-bearing loading in multi-megawatt machines, but often with aims different from characterising time-varying
load behaviour. For example, valuable studies have been undertaken which explore: impacts of including bedplate stiffness in
models of four-point support geared machines (Wang et al., 2020), the impacts of yaw error on main-bearing fatigue damage
and lifetime (Cardaun et al., 2019), the influence of elastic surroundings, clearance and applied load direction/magnitude on
load distributions within the downwind (in this case, cylindrical roller) portion of a four-point mounted drivetrain support
(Kock et al., 2019). There are also studies in the literature which relate more directly to the focus of the current paper. In
Zheng et al. (2018, 2020a, b) a quasi-static contact model was developed for analysis of double-row TRBs, including effects
of load, angular misalignment and friction. This model was applied to investigate internal load and life effects on bearings
found in large (5MW) direct-drive machines, including the influence of oscillating load and rotational speed on main-bearing
fatigue life. While this work provides useful insights, applied load variations were simplified to sinusoidal in nature (for a range
of amplitudes) and thrust was assumed constant. The applied load variations were therefore not representative of real world
main-bearing operation.

To-date, there have been no studies focused on the general time-varying characteristics of main-bearing loading in multi-
MW wind turbines. Investigation of the characteristics, relative magnitudes and drivers of such loads would generate important
insights into main-bearing operating conditions, while also providing load case inputs for future work. Most existing studies
in this area utilise high complexity models, requiring specific and often proprietary information regarding system components
and geometry. As in previous research on smaller geared turbines, simpler analytical representations would allow for fast
and efficient estimation of loading for a large number of cases, without requiring sensitive information. Where appropriate
higher fidelity models are available, more detailed analyses could then be undertaken for key load cases. Furthermore, for





some applications (e.g. controller evaluation) analytical model outputs may provide as much information as is required. As
stated earlier, the focus of this work is therefore *the application of simplified main-bearing load response models to investigate
characteristics and key drivers of main-bearing loading in large (5-10 MW) wind turbines.*

## 2.2 Wind turbine aeroelastic modelling and upscaling

The aeroelastic software used in this study is DNV-GL Bladed, an industry standard blade element momentum modelling tool
certified for wind turbine design. The models within DNV-GL Bladed include structural representations of the blades, tower,
and nacelle with full coupling of these sub-components using multi-body methods. Outputted loads from simulations include
effects of aerodynamic loading, inertial response of structural components, tower shadow effects, and gravitational loading
(including effects of drivetrain tilt). For the research presented here, the output quantities of interest are the three force and
three moment components at the hub, which we shall refer to as "hub loads".

In the current study we analyse the "baseline" NREL 5MW wind turbine model (Jonkman et al., 2009) and two upscaled
wind turbine models with rated powers of 7.5 and 10 MW (Thompson, 2018). The two larger models are obtained using
similarity scaling to isolate size-related impacts on bearing loading and response without modification of design specifications.
The scaling process assumes that key nondimensional parameters such as rotor power and thrust coefficients are the same for
the three wind turbines under the assumption that the rotor Reynolds numbers are above the transitional Reynolds number to
fully developed turbulence over the blades, rotor and wake. Given the extremely high Reynolds numbers of the three modeled
turbines, both for the rotor and the individual blades, this is almost certainly the case (Vijayakumar and Brasseur, 2019). In
the scaling, tip speed ratio is held constant so as to maintain the same power density (power/area) for the three rotors. Power
therefore scales strictly on rotor radius. Assuming similarity scaling over the three turbines, rotor mass and inertia scale on the
third and fifth power of rotor radius, respectively. In practice, mass and inertia will likely scale somewhat differently due to
technological adjustments made with increasing turbine size (Jamieson and Hassan, 2011). Such potential adjustments are not
considered here. With load and material properties held constant, the stiffness of the wind turbine structure scales linearly with
rotor radius.

## 2.3 Wind field modelling

Wind turbine rotors interact with large areas of turbulent wind, this being the driver for both power generation and loading
on components which may contribute to premature main-bearing failures. Turbulent wind fields are complex and difficult
to model accurately. Simplified kinematic representations[1] are therefore often used as a proxy when performing design and
analysis tasks. These kinematic wind turbulence models treat the wind field as consisting of deterministic components[2] (hub-
height mean wind speed, shear profile, yaw offsets and veer) overlaid with coherent stochastic variations (Mann, 1998; Kelley
and Jonkman, 2005). In reality, wind fields contain coherent atmospheric turbulence eddies with stochastic characteristics

---

[1]Such representations ensure generated wind fields have appropriate second order statistics (based on measured wind data), but, are unable to recreate true
turbulence eddy structure.

[2]For a more detailed description of these components see Hart et al. (2020).





that create responses at the lower-frequency large-scale end of spectra. However, interactions with rotating blades create both
high and low frequency response to the passage of quasi-deterministic coherent eddy structure (Lavely, 2017; Nandi et al.,
2017). Irrespective of how wind-field and turbine modelling is undertaken, the presence of turbulence necessitates statistical
analysis of large numbers of outputs to produce converged statistics. While important for making design based decisions, such
additional complexity is not necessarily of benefit at earlier stages of analysis in which load drivers and general trends are
sought. This logic is similar to that of Gould and Burris (2016), in which effects of shear on gearbox loads were considered. In
the current study we therefore consider response to steady mean velocity profiles, in which shear and veer are present, and do
not consider the impacts of turbulence fluctuations on main-bearing response. These wind fields will be referred to as "steady
mean wind fields" in this paper. In addition, we will consider main-bearing response to simple deterministic gusts and wind
direction change events, which will likely contain elements of turbulence eddy response (see Section 3.2 and Appendix A).

## 2.4 Wind turbine control

It is essential to consider the turbine controller when discussing wind turbine dynamics. Modern multi-megawatt wind turbines
are almost exclusively variable-speed, pitch-regulated machines. A well designed controller[3] will follow the desired opera-
tional strategy[4], have a bandwidth of at least 1 rad/s because the main turbulent structures in the wind are contained within this
frequency range (Burton et al., 2001), minimise loads on structural components, account for aerodynamic non-linearities, and
minimise actuator activity. To address controller requirements, advanced control methods have been developed to minimise
blade loads (Bossanyi, 2003; Leithead and Stock, 2016), dampen tower vibrations (Leithead and Dominguez, 2006; Chat-
zopoulos, 2011; Leithead and Stock, 2016), and increase power capture in certain conditions (Pedersen et al., 2020). Whilst
some of these goals may be complementary, often they are competing demands. Due to the wide variety of goals combined
with the wide variety of possible implementations, two wind turbines that are ostensibly quite similar may have controllers that
are highly dissimilar. It is therefore important that, when comparing variations in non-control parameters, similarly designed
controllers are used. While operating at below-rated power, wind turbine controllers use torque demand as the main control
action. Once rated power is reached, blade pitch angle becomes the main control action. Rated power is normally reached at
a wind speed of 11-12m/s. Torque can be altered very quickly (with a time constant of a tenth of a second or so) whereas
changes in pitch are an order of magnitude slower. For reference, the NREL 5MW model power and thrust operational strategy
curves are shown in Figure 1. To the best of the authors' knowledge, the effect of the controller on main-bearing loads has not
previously been presented in the literature.

---

[3]Note that "design of a controller" includes the controller architecture and implementation, as well as the tuning of the controller itself.

[4]"Operational strategy" here refers to the operating points a wind turbine is designed to move through as conditions vary. A wind turbine controller is
therefore employed to attempt to ensure a turbine operates according to its operational strategy as far as is possible/reasonable. For example, Figure 1 shows
the designed operating strategy with respect to power and thrust versus wind speed for the 5 MW wind turbine.





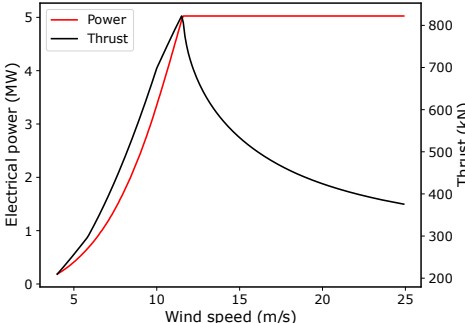

**Figure 1.** Power and thrust operational strategy curves for the NREL 5MW wind turbine model used in this study.

## 3    Methodology

Analysis of main-bearing load characteristics was undertaken in two stages. First, a range of aeroelastic simulations were run for the 5, 7.5 and 10 MW turbine models. Second, hub loads were extracted from these simulations and input into simplified main-bearing representations. Main-bearing loads were analysed similarly to previous work (Hart, 2020). Details of these analysis stages will now be provided, including the simplified main-bearing response models for two main-bearing configurations.

### 3.1    Control considerations for the current study

When considering fundamental loading mechanisms for main-bearings it is necessary to ensure a consistent control approach is used across models, and that the controller is kept basic to avoid additional interactions and effects. However, for upscaled turbine models, consistency of the control approach becomes difficult due to inevitable changes in bandwidth in the absence of advanced control techniques. It was therefore concluded that influences of controller dynamics should be minimised for the larger turbines. Hence, the upscaled turbine models (7.5 and 10 MW) were only used in steady mean wind field cases, with control dynamics removed completely by maintaining torque and pitch at appropriate equilibrium operating points after initial transients from simulation start-ups had settled. Both the controller and the turbine model of the NREL 5 MW turbine are considered representative; therefore, it was deemed relevant to still consider loading and control interactions in transient cases for this turbine model. Analysis using the 5 MW model was therefore undertaken in both steady mean wind fields (while removing control dynamics, as for the larger models) and non-steady wind fields (with the controller active). The controllers used throughout were the "basic controllers" defined in Thompson (2018).

### 3.2    Turbine models and simulations

The baseline wind turbine model used in this study is the NREL 5 MW (Jonkman et al., 2009) modelled in DNV-GL Bladed. This study also considers larger 7.5 MW and 10 MW wind turbine models upscaled from the 5 MW model as described in Section 2.2. Due to the high complexities associated with atmospheric turbulence, it was concluded the current study should focus on the effect of non-stochastic wind cases, including steady mean wind fields and deterministic non-steady events such



as gusts, on main-bearing loads. As motivated by considerations outlined in Section 2.4, for the steady mean wind cases wind field effects alone on main-bearing load response were isolated by holding control variables fixed. In the case of non-steady
events, the control system necessarily remained active so the controller will influence main-bearing loads.

**Steady mean wind fields:** The first set of analyses considered main-bearing loading resulting from simulations in steady wind fields with specified values of mean wind speed (at 90m), power-law shear exponent, yaw offset, and veer gradient. All turbines were subjected to identical wind fields, irrespective of turbine size. The analysis therefore provides insight into impacts of turbine selection at a given site. Table 1 summarises the steady mean wind cases which were analysed. The mean
wind speed range is that normally associated with wind turbine operation. Similarly, all shear and veer values used here have been observed in measured site data (Murphy et al., 2020). During normal operation, the yaw angle of the nacelle relative to the mean wind inflow direction is controlled to be kept within approximately $\pm 8°$, with this allowable yaw error a trade off between ensuring maximum energy capture while minimising yaw duty. Recent advances in wind farm control suggest benefits from wake steering controllers, with yaw angles for wake steering purposes reaching 30 or even $40°$ (Fleming et al., 2017).

| Case | Ref. height mean wind speed ($v$) [m/s] | Shear exp. ($\alpha$) [-] | Yaw offset ($\varphi$) [°] | Veer gradient ($\gamma$) [°/$m$] |
|---|---|---|---|---|
| A | 4, 6, 8, . . . , 22, 24 | 0.2 | 0 | 0 |
| B | 10 | 0.1, 0.2, 0.3, 0.4, 0.5 | 0 | 0 |
| C | 10 | 0.2 | -20, -12, 0, 12, 20 | 0 |
| D | 10 | 0.2 | 0 | -0.6, -0.2, 0, 0.2, 0.6 |

**Table 1.** Simulated steady mean wind cases. The mean wind speed reference height is 90m, with power law shear profiles also referenced to this height.

**Deterministic non-steady events:** Because the controller for the 5 MW model is considered representative, dynamic response to deterministic events for that model were also investigated. This included both gusts and extreme wind direction changes as defined by IEC standards (IEC, 2005-08). Results will be presented for a class 1A gust during operation at 8, 11 and 14m/s (at 90m) and extreme wind direction changes ($30°$ in 6 seconds) during operation at 8 and 16m/s. In each case shear was present, with the shear exponent set to 0.2. Formal definitions of simulated non-steady events are provided in Appendix A.

### 3.3 Main-bearing configurations and analytical modelling

This study will focus on two hub and main-bearing configurations. The first configuration (referred to here as a *centered support*) features two main-bearing rows widely spaced on either side of the hub, typically transferring the non-torque loads to a stationary internal mounting or extended frame. This configuration is representative of the Alstom ECO100 geared (Guo et al., 2014, 2015) and the GE Haliade direct-drive (GE, 2021) turbines. The second configuration (referred to here as an
*overhung support*) features two main-bearing rows more closely spaced together both downwind of the hub, transferring the non-torque loads in some cases to a stationary internal frame representative of many newer direct-drive turbines (Gaertner et al., 2020) while in others transferring them to a bearing housing representative of many geared turbines (Demtröder et al., 2019).



Examples of such configurations are depicted in Figure 2 for the case where the hub rotates around a stationary mounting. This is illustrative only, as the model outlined below applies equally in cases where the rotor drives an internal shaft. For the sake of

simplicity, the blades and rest of the drivetrain (generator and/or gearbox) are not depicted.

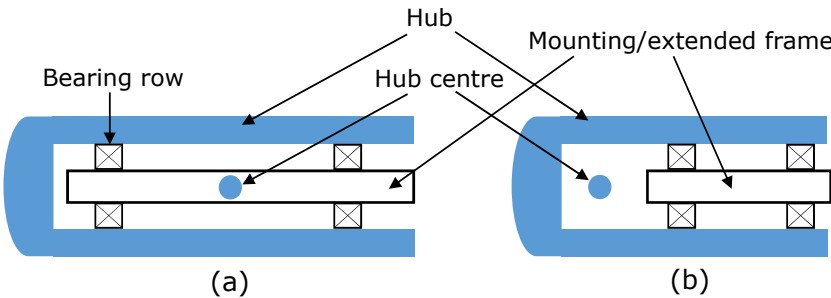

**Figure 2.** Hub and main-bearing configurations studied in the current paper. Throughout the paper, these will be referred to as (a) centered support, and (b) overhung support.

Both rotor support configurations may be described using the general representation shown in Figure 3. The six degree-of-freedom hub loads, $F_h$ and $M_h$, act at the hub center, which is located a distance of $L_h$ (positive upwind) from the midpoint of the main-bearing rows. Bearing row spacing is given by $2L_b$. Shaft, generator, and/or gearbox weight are neglected in the current analysis as they are often much less than the rotor weight, and static rather than dynamic loads of interest to this work.

If known, they could easily be incorporated into the modelling framework presented here. The hub, driveshafts, and frame are approximated as rigid. The system is statically indeterminate (Stirling et al., 2021). However for the study described herein, the bearing rows are spaced a reasonable distance apart. As such, load response at each row is expected to be dominated by forces, with contributions from moments acting across individual bearing rows likely small in comparison (Tong and Hong, 2014; Zhang et al., 2019). In the current study, load response is therefore approximated as consisting of only three degree-of-freedom

forces, $F_1$ and $F_2$, at individual bearing rows. Doing so yields a statically determinant system. In wind turbines, force dominant response is often achieved by selecting roller contact angles such that a higher "effective length" is achieved than the physical distance between bearing rows (i.e. TRBs in an O-configuration). All axial loads are assumed to be reacted by only one bearing row. If the axial load direction changes ($F_h^x > 0 \rightarrow F_h^x < 0$) then axial support may switch to the other row depending on the bearing design.

Note, while it has been argued that moments acting across individual rows may be neglected in the context of estimating full system response (due to their expected contribution relative to system forces), this does not necessarily imply that individual-row moment reactions present in reality are irrelevant to bearing health and reliability. A bearing's sensitivity to moment driven differential roller loading will ultimately dictate whether even small levels of moment loading across bearing rows may compromise bearing integrity. But, because the evaluation of moment responses would require more detailed main-bearing

information and modelling, the present study focuses on characteristics and drivers of estimated row force responses only.



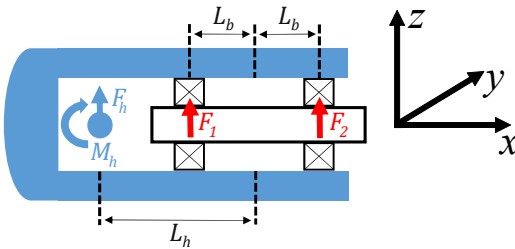

**Figure 3.** General representation of modeled main-bearings and DNV-GL Bladed reference frame. Applied loads, $F_h$ and $M_h$, act on the hub at its centre. Main-bearing row force reactions, $F_1$ and $F_2$, are also shown.

With the above approximations in place, force responses at each bearing row for the generic drivetrain shown in Figure 3 are given by,

$$F_i^x = -\delta_{ij} F_h^x \quad (i = 1, 2.\ j \text{ indicates the thrust supporting row}), \tag{1}$$

$$F_1^y = -\frac{1}{2}\left(\left(1 + \frac{L_h}{L_b}\right) F_h^y - \frac{M_h^z}{L_b}\right), \quad F_2^y = -\frac{1}{2}\left(\left(1 - \frac{L_h}{L_b}\right) F_h^y + \frac{M_h^z}{L_b}\right), \tag{2}$$

$$F_1^z = -\frac{1}{2}\left(\left(1 + \frac{L_h}{L_b}\right) F_h^z + \frac{M_h^y}{L_b}\right), \quad F_2^z = -\frac{1}{2}\left(\left(1 - \frac{L_h}{L_b}\right) F_h^z - \frac{M_h^y}{L_b}\right), \tag{3}$$

where $\delta_{ij}$ is the Kronecker delta.

Having derived equations which estimate row force responses in overhung and centered support configurations, it remains to specify the parameters $L_h$ and $L_b$ for the 5, 7.5 and 10 MW wind turbine models. The rotor weight moment in overhung wind turbines is a significant load. As such, this value is minimised wherever possible and, hence, as turbines become larger main shafts and bearings tend to increase in diameter as opposed to the drivetrain becoming longer; sometimes larger machines will even see a reduction in the drivetrain length. For these reasons, in the current study it was assumed that lengths, $L_h$ and $L_b$, remain the same across the three turbines. For the centered support, $L_h = 0$m and $L_b = 2$m. For the overhung support, $L_h = 2$m and $L_b = 0.2$m (one order of magnitude less than $L_h$). These parameters were estimated from available schematics of such drivetrains and are intended to be representative, while not necessarily exact. The simplicity of identified equations allows the sensitivity of results to drivetrain dimensions, along with alterations in outputs resulting from differing values, to be readily understood. Since the presented equations for estimating force response rely on the assertion that "bearing rows are spaced a reasonable distance apart", it follows that the quality of these force estimates will vary with $L_b$. Appendix B considers such questions in detail, via an alternative derivation of main-bearing load response. It is shown there that radial force estimates (Equations 2 and 3) remain viable for the considered overhung support (where $L_b$ is smaller), albeit providing a poorer estimate than in the centered support case.

Models of this type have previously been shown to be able to reproduce main-bearing load reactions calculated using higher fidelity finite element representations of a shaft/main-bearing system (Stirling et al., 2021). Depending on model specifics, mean percentage errors of between 1.54% and 22.74% were reported. While this provides some confidence in the approach taken here, an exact level of accuracy in the current case is not yet known. The presented models should therefore be interpreted





as providing first order engineering estimates of load response for the main-bearings in question. Inline with the stated aims
of this work, such models allow for the characteristics, drivers, and general orders of magnitude of main-bearing loading to be
investigated for these machines. As such, they are sufficient for the work undertaken in the current paper. Model limitations
should, however, be kept in mind when interpreting results.

## 4    Results

Main-bearing loads for steady mean and non-steady analyses will now be presented. In order to make interpretation of results
more intuitive, bearing support reactions will be presented as the *load applied to the bearing row*; this being equal in magnitude
to the *bearing reaction* shown in Figure 3, but opposite in direction. Positive directions in this adjusted frame are taken as
vertically upwards and horizontally to the right when viewed from upwind. These adjustments result in the plotted applied
forces (acting on bearing rows) being aligned with the true direction of acting forces when looking at the turbine rotor from
upwind. Force results in this adjusted frame will be denoted by $\tilde{F}_1$ and $\tilde{F}_2$ for the centered support and $\tilde{F}_{0,1}$ and $\tilde{F}_{0,2}$ for the
overhung support.

### 4.1    5 MW model results: steady mean wind fields

The effects of deterministic wind field parameters, including reference-height mean wind speed, shear exponent, yaw angle and
veer described in Section 3.2, on main-bearing load response for the 5MW wind turbine are examined in this section. Under
steady operation these structures result in repeating load patterns (at "3P", *i.e.* three repeated loops occur per rotation of the
turbine rotor) for each bearing row (Hart, 2020). Figure 4 shows the repeating force patterns for each row in the overhung and
centered hub-main-bearing support configurations. Overhung support results for upwind and downwind rows are **orange** and
**green**, respectively. Centered support results for upwind and downwind rows are **red** and **black**, respectively. Subplots 4a-4d
show the results for varying wind speed ($v$), shear exponent ($\alpha$), yaw offset ($\varphi$) and veer ($\gamma$) respectively. Parameter values
for the overhung support case are indicated, while those for the centered support are omitted for clarity. Overhung support
loads are significantly larger, in terms of both mean load level and the magnitude of load fluctuations, than in the centered
support loads. This is not surprising because rotor moments must be reacted by larger forces acting at smaller distances for the
overhung support. In order to contextualise these results, it is useful to identify a reference load level with which to compare.
In the absence of operationally induced loads, the main-bearing must still support rotor weight. Being distributed across two
rows, reference force loading for an individual row is therefore taken to be half the rotor weight, $W$. For the 5MW model,
$W/2 = 544$ kN. For the overhung support, rotor weight also generates a moment which, for the current model, requires a
nominal response of closer to $10 \times W/2$. But, for the sake of simplicity a single reference load level of $W/2$ will be used when
discussing results. The mentioned factor of $\sim 10$ should therefore be kept in mind when interpreting results.

The looped load structures identified in previous work are clearly visible in these results. For both wind speed and shear
exponent variations, load fluctuations are small when $v$ or $\alpha$ are small and steadily increase as these variables increase (*i.e.*
the loops become bigger). As either wind speed or shear exponent values increase, both main-bearing configurations also see



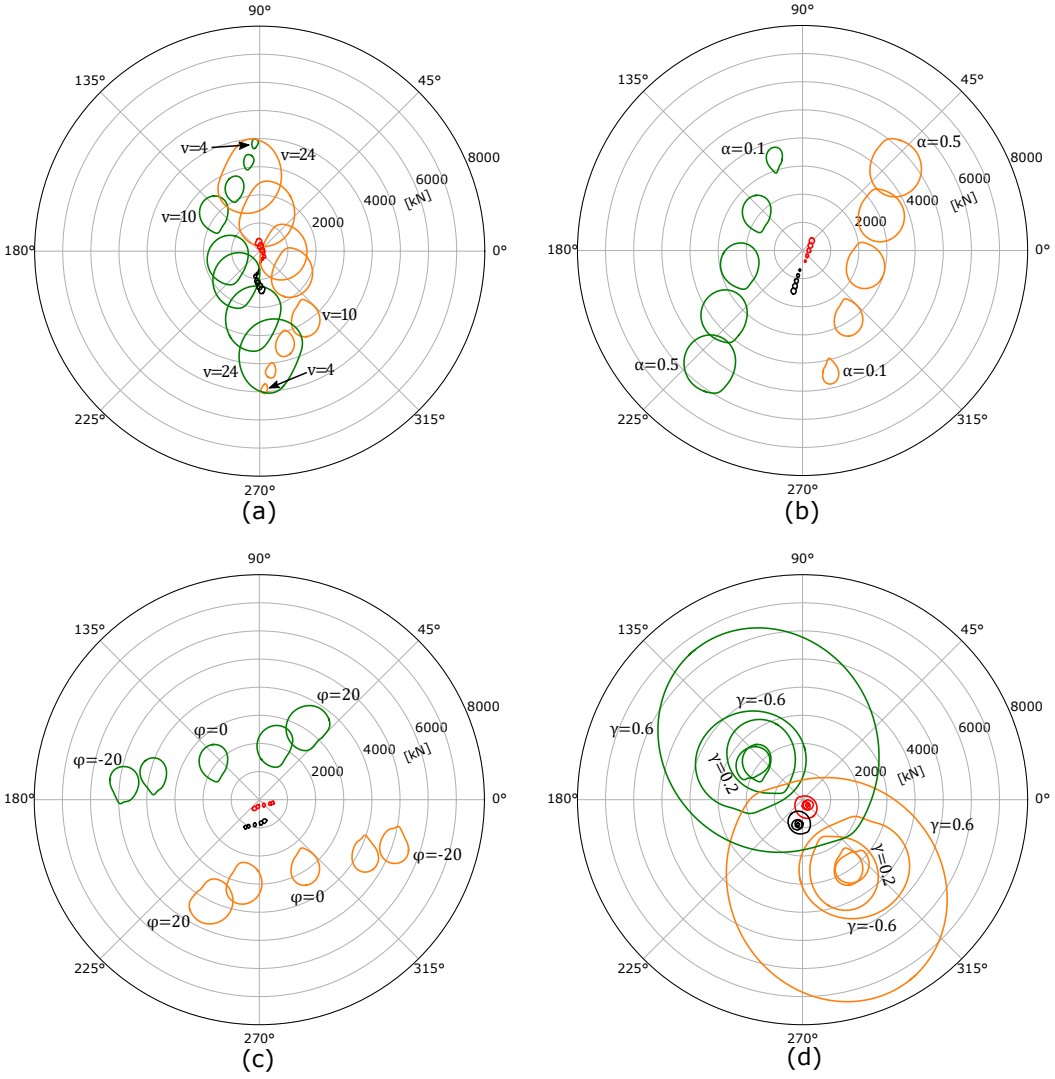

**Figure 4.** Bearing row radial force response patterns for the 5MW turbine under various values of reference-height wind speed ($v$), shear exponent ($\alpha$), yaw offset ($\varphi$) and veer ($\gamma$) are shown in subplots 4a-4d, respectively. Overhung support results for upwind and downwind rows are **orange** and **green** and centered support results for upwind and downwind rows are **red** and **black**, respectively.

corresponding changes in mean load levels (*i.e.* the distance from the loop centre to the origin). The directions of force reactions are also affected. The change in the trend of the wind speed results evident in Figure 4a is driven by the turbine operating strategy, specifically thrust levels reducing once rated power is reached (see Figure 1). Some conditions lead to fluctuations

passing close to (or even through) the origin, such cases would manifest as repeated loading and unloading of radial forces on that bearing row. Yaw angles primarily change the mean load level, with load fluctuation magnitudes significantly less affected. In agreement with results reported in the literature (Cardaun et al. (2019), who considered fatigue life impacts of yaw offsets)



the direction of yaw has an asymmetrical effect on row force response with respect to both mean load levels and loop sizes. In part because load results for $\varphi = 0$ are offset from $\tilde{F}_y = 0$, the average size of yaw related forces are highest for negative

$\varphi$, although the amplitudes of resulting fluctuations are lower. The offset itself, at $\varphi = 0$, results from rotor tilt which acts to slightly increase the effective wind speed on one side of the rotor while reducing it by a similar amount on the other. Veer, on the other hand, strongly impacts the magnitude of load fluctuations, while leaving mean load levels relatively unchanged. Similar to wind speed variation results, certain veer conditions will drive unloading/re-loading events of significant magnitude. The highest magnitudes of row force fluctuations occur for the largest $\gamma$ magnitudes (positive or negative), with $\gamma = 0.6$ resulting

in force fluctuations of close to $15\times$ and $2\times$ the reference load level ($W/2$). In terms of mean loading, variations in wind speed and shear exponent lead to loop centre locations being shifted by close to $14\times$ and $2\times$ the reference load level when considering full ranges of variable values. The turbine operating strategy (specifically the switch to above rated operation) was already seen to influence load behaviour in Figure 4a. Trends for other parameters are also affected by this switch; for example, radial loads under yawed inflow at at an above-rated wind speed of $v = 16$m/s, as illustrated in Figure 5. Whereas at 10m/s

yaw angles induced load variations mostly horizontally, in above-rated operation the dominant effect is to induce variations vertically.

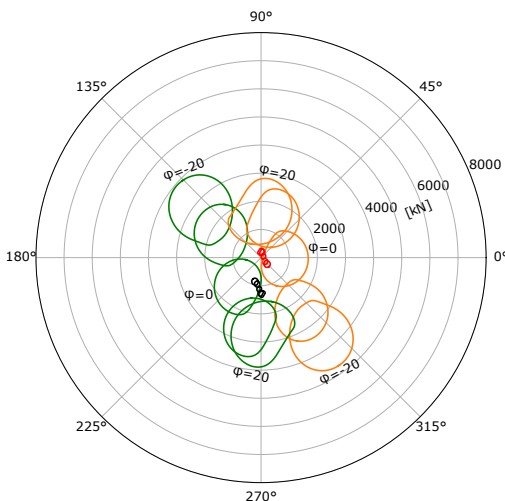

**Figure 5.** Bearing row radial force response patterns (for the 5MW model) under various values of yaw offset ($\varphi$) with a reference-height mean wind speed of 16m/s. Overhung support results for upwind and downwind rows are **orange** and **green** and centered support results for upwind and downwind rows are **red** and **black**, respectively.

Axial load fluctuations on the thrust-supporting row of the main-bearing occur simultaneously with fluctuations in its radial loading, driven by the designed aerodynamic thrust operating strategy (Figure 1) with wind speed. The thrust strategy, and so the resulting axial loads on the main-bearing, also interacts with other wind field parameters. For example, with respect to

the turbines operating strategy a yaw offset is akin to a reduction in wind speed and, hence, turbine thrust changes under yaw may be inferred. Table 2 shows this quantitatively, listing average values of key operational variables at $v = 10$, 16m/s and





$\varphi = 0,\ 20°$. During below rated operation, the introduction of a yaw offset means the turbine moves to a point of lower torque and lower thrust (reductions of around 4% and 5%, respectively). In above-rated operation, introduction of the same yaw offset leads to a reduction in pitch angle and an increase in thrust (changes of around -12% and +3%, respectively). These changes in thrust/axial bearing loading under yawed flow occur alongside the illustrated changes in radial loading. Characteristics of applied radial and axial loading on the main-bearing are therefore linked and, importantly, both are driven by the turbine's operational strategy. It should, however, be emphasised that each is influenced by different aspects of the inflow and rotor loading, with thrust driven by the *summation* of forces across the rotor and radial main-bearing loading strongly driven by moments generated from the *distribution* of forces across the rotor (Lavely, 2017; Hart, 2020). Therefore, while a strong link is present the relationship between the two is not straightforward.

| Wind speed (m/s) | Yaw offset (deg.) | Aero. torque (kNm) | Pitch angle (deg.) | Thrust (kN) |
|---|---|---|---|---|
| 10 | 0 | 2886 | 0 | 698 |
| 10 | 20 | 2773 | 0 | 665 |
| 16 | 0 | 4256 | 11.9 | 490 |
| 16 | 20 | 4256 | 10.5 | 503 |

**Table 2.** Mean values of key operational variables for the 5MW NREL turbine during steady operation at specified values of mean wind speed and yaw offset.

An important aspect of main-bearing fatigue life determination per ISO 281 is the ratio of axial to radial loading. Different load factors are chosen based on this ratio as compared to a "limiting value", $e$, based on the bearing contact angle. Additionally as discussed in previous work (Hart et al., 2019), relative levels of axial vs radial loading may also have important implications for bearing health, with some combinations increasing the risk of roller unseating, skidding, or skewing. Relationships between fluctuations in thrust and main-bearing radial forces were therefore considered. It was found that the radial force response in overhung and centered support configurations can have different relationships with thrust, with inflow characteristics influencing these differences. Figure 6 illustrates these findings using three example steady mean wind cases. In order to consider correlational aspects of load relationships, standardised time-series of thrust and radial force magnitude[5] are presented. As was the case for radial loads, thrust fluctuations occur at 3P. These results clearly show that, for both main-bearing configurations, wind field conditions exist in which radial force magnitudes vary in-phase with thrust loads (*i.e.* the maxima and minima of the signals coincide), but, conditions also exist in which these variations are out-of-phase with thrust (*i.e.* one reaches a maxima while the other reaches a minima). Intermediate phase relationships (not shown here) between these extremes also occur. It is interesting to note, in addition, that differences between the overhung and centered support thrust vs force relationships are not static, but also dependent on wind field characteristics. Implications of these finding for bearing operation and internal conditions are not yet clear, but, presented results demonstrate again that main-bearing thrust/axial load vs radial load relationships are non-trivial and so should necessarily be considered as part of ongoing research in this area.

---

[5]Explicitly, for a time-series signal $f_t$ (for $t \in T$) the standardised signal is $(f_t - \mu_f)/\sigma_f$ (for $t \in T$) where $\mu_f$ and $\sigma_f$ are the mean and standard deviation of $f_t$ over the interval $T$. The resulting standardised signal has a mean of 0 and standard deviation of 1 by construction.

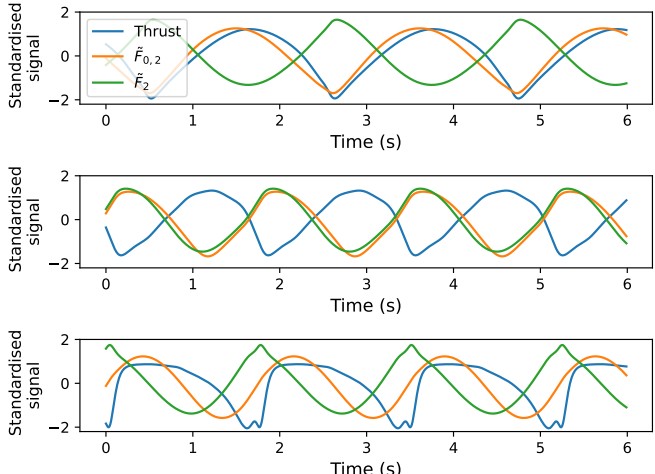

**Figure 6.** Time-series of thrust and main-bearing row force response magnitudes. All outputs are presented as standardised signals, allowing for easy consideration of correlations. From top to bottom, wind field characteristics are: $v = 8$ m/s and $\alpha = 0.2$, $v = 18$ m/s and $\alpha = 0.2$ and, finally, $v = 10$ m/s, $\alpha = 0.2$ and $\varphi = -20°$.

## 4.2 Upscaling results: steady mean wind fields

Results obtained from simulating upscaled turbines (7.5 and 10 MW) in identical wind fields to those used for the 5 MW turbine will now be presented. With respect to the posed research question, an important consideration here is whether response behaviour for the larger machines deviates from that of the 5 MW turbine. Reference radial force loading for individual main-bearing rows was previously taken to be half the rotor weight. Because the three turbine models differ in terms of their physical size, 'normal' loading is different for each. As such, load responses should be considered *relative* to an appropriate reference load in each case. Force results for all turbines were therefore non-dimensionalised using half of their respective rotor weights in each case. In order to consider load variability, ellipses were fitted to the identified load loops, with elliptical areas then calculated (Hart, 2020). Load loop area results were non-dimensionalised using the square of the W/2 value of each turbine. Recall that on upscaled models the rotor weights scale cubically with the turbine rotor radius (see Section 2.2); therefore, overlapping lines in dimensionless results plots for the three models indicate that main-bearing loads are also increasing cubically with the rotor radius.

### 4.2.1 Load loop centre results

As discussed in previous sections, the radial load magnitude[6] provides an indication of mean loading associated with a given operating point. By expressing loop centres as a combination of (non-dimensionalised) magnitude and direction, results from varying a given parameter may be summarised in a single figure for all three turbine models. Figure 7 shows these results on the

---

[6]Specifically, the magnitude of the centre point of the ellipse fitted to identified loops in radial load trajectories.




downwind main-bearing rows of both main-bearing configurations for all three turbines for variations in the reference height wind speed. Overall, the behaviour and trends described for the 5 MW model in Section 4.1 also hold for the larger turbines. In

particular, the chosen reference load values are justified by these results, because non-dimensionalised load magnitudes for the three models are in close agreement. Importantly, overlapping non-dimensionalised results (discussed at the beginning of this subsection) for the three turbines indicate that mean loading during steady operation is scaling cubically with the turbine rotor radius for turbines operating in identical wind fields. Similar conclusions hold as the other wind characteristics are varied, with some instances of faster than cubic scaling apparent. Dimensionless yaw results, Figure 8, show that yaw related asymmetries

in mean radial loading for the overhung support persist in the larger turbines, but, with increases in mean radial loading at negative yaw becoming more pronounced as turbine size increases. Shear and veer results may be found in Appendix C.

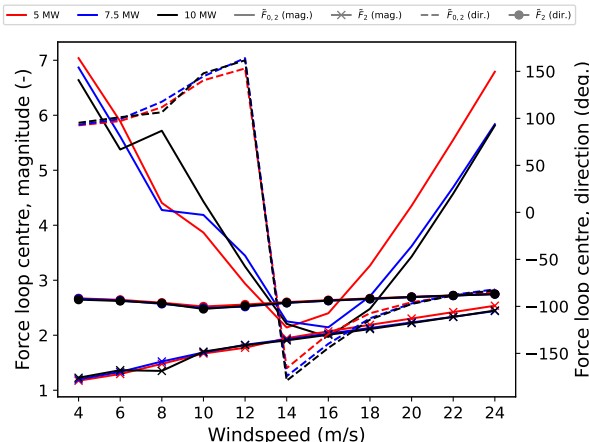

**Figure 7.** Non-dimensionalised, downwind row radial force-loop-centre magnitude and direction values for overhung and centered main-bearing configurations across the three modelled turbines as the reference height wind speed is varied.

#### 4.2.2   Load loop area results

Radial load loop areas[7] capture the load variability (with respect to both magnitude and direction) at each operating point. Non-dimensionalised radial load loop area results for variations in wind speed and veer gradient on downwind rows are given

in Figures 9 and 10. Centered main-bearing row results have been scaled by a factor of 50 to ensure trend features are visible on a single plot. Wind speed results show close agreement in terms of (dimensionless) load variability values across the three turbine scales. Similar to mean load results, this indicates that radial load fluctuation magnitudes are scaling cubically with turbine rotor radius for operation in identical wind fields. Faster than cubic scaling can be observed in the dimensionless loop area results for veer, Figure 10. Consistent with 5 MW model results, large veer gradients were found to lead to the highest

---

[7]This being the area of the best-fitting ellipse.

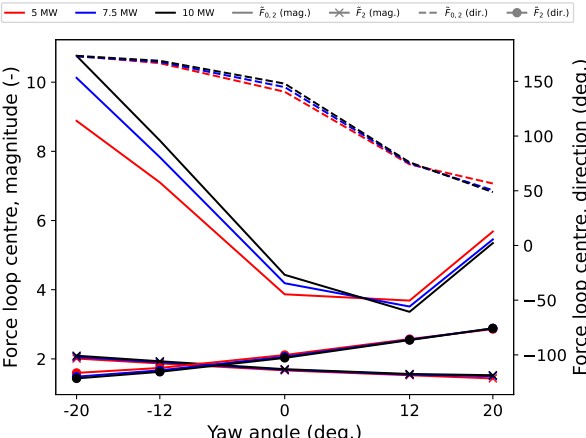

**Figure 8.** Non-dimensionalised, downwind row radial force-loop-centre magnitude and direction values for overhung and centered main-bearing configurations across the three modelled turbines as the yaw offset is varied.

magnitude radial load fluctuations across all modelled turbine scales. In addition, veer has been found to drive the nastiest scaling behaviour as turbine size increases. Shear and yaw results are provided in Appendix C.

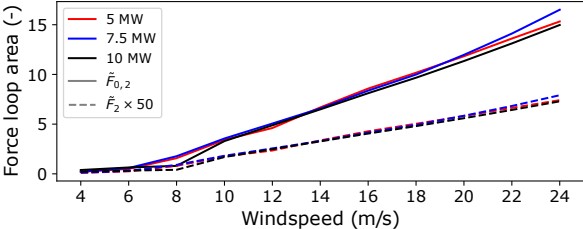

**Figure 9.** Non-dimensionalised downwind row radial load-loop-area results at various reference height wind speeds for overhung and centered main-bearing configurations across the three modelled turbines.

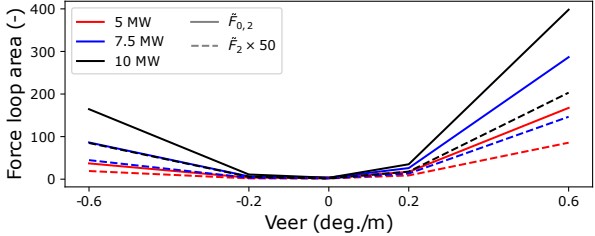

**Figure 10.** Non-dimensionalised downwind row radial load-loop-area results for various veer gradients for overhung and centered main-bearing configurations across the three modelled turbines.





### 4.3 5MW model results: deterministic non-steady events

Main-bearing load response for the 5MW turbine during non-steady events, with control active (see Section 3.1), are now presented. For the sake of brevity, only the load response in the upwind row of the overhung support configuration will be
shown. Gusts are considered first. Figure 11 shows standardised wind speed values for each gust, *i.e.* the figure gives the shape of wind speed variations occurring during gust events. Gusts applied during simulations are scaled versions of this curve centred at the appropriate nominal wind speed. The vertical axis is the number of standard deviations away from the mean occurring at each point in time. See Appendix A for the explicit gust equation.

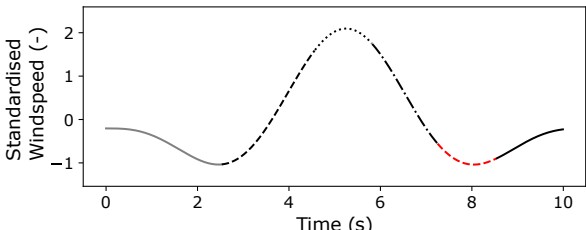

**Figure 11.** Standardised values of the wind speed (at 90m) during simulated gust events. Vertical axis values give the number of standard deviations from the mean. Line segments in this figure correspond to those in Figure 12.

Figure 12 shows axial (i.e. thrust) and radial responses due to gusts centred on $v = 8$m/s (subfigures a-c), 11m/s (subfigures
d and e) and 14m/s (subfigures f-h). The turbine's thrust operating strategy curve is also provided for reference. The start point of each load trajectory is indicated by a red star. At $v = 8$m/s (Fig. 12a-c) torque (the below-rated control variable) varies in response to the gust as the turbine attempts to maintain optimal efficiency. As stated in Section 2.4, torque provides a comparatively fast control action, meaning axial loading follows the design thrust curve relatively closely during the event. Radial force fluctuations are small relative to the mean load level, with the load direction also remaining fairly constant
throughout. At $v = 14$m/s (Fig. 12f-h), however, load response is somewhat more dramatic. Pitch (the above-rated control variable) is a much slower control action, as evidenced by the lag which may be seen when comparing pitch values to the gust itself[8]. For example, wind speed peaks at about 5.5 seconds, while the peak in pitch angle does not fall until around 7 seconds (see Figure 12h). A certain amount of "overshoot" in turbine response would therefore be expected to take place. Axial bearing loading during the gust deviates significantly from the design operating curve. A short initial drop when wind speed
decreases is followed by a fast rise to a local peak of just below 1000kN as the gust itself peaks. Then, as pitch values reach their maximum, the main-bearing becomes rapidly axially unloaded (even seeing some small negative values). This fast drop in axial loading is understood to be initially caused by the rapid drop in wind speed occurring at a large pitch angle. This causes

---

[8]A more detailed description of lag effects is as follows: when a change in wind speed occurs there is a lag before the resulting change in torque reduces the generator speed due to inertia in the aero-elastic dynamics of the wind turbine. There is a further lag between a reduction in generator speed and a change in pitch angle, due to time constants associated with the controller and pitch actuators. Additional lag occurs between pitch and changes in aerodynamic torque due to induction-lag effects. The combined effect of these lags is ultimately what results in the dramatic load sweeps observed in Figure 12d-g when pitch is active.





the turbine to "rock" forwards and then back again, which in turn leads to a brief period of zero/negative axial loading. Axial forces then climb rapidly to the overall maximum value seen during the gust. Radial forces during this gust also see significant

variations in both magnitude and direction. Steady operation load loops are visible initially (the grey line segment), followed by more chaotic variations as the gust progresses. Lags in pitch angle values relative to wind speed result in some of the highest forces occurring after the gust has peaked. From around 6 seconds, radial loads see significant variations in both magnitude and direction, with radial forces acting to lift the rotor and then rapidly drop it. As with axial loading, the largest magnitude radial load is seen towards the end of the gust. At this point, while the wind speed has returned to its nominal level, the pitch

angle can be see to lie far from its steady operational value. This is a concern, because it indicates that a second gust hitting the turbine shortly after the first would elicit an even more severe load response.

At $v = 11\mathrm{m/s}$, the turbine is initially in below-rated (torque controlled) operation, it then switches to above-rated (pitch controlled) operation between roughly 4.5-8.5s, before torque control is resumed. This is reflected in the load response, with axial bearing loading at $v = 11\mathrm{m/s}$ initially behaving as during the $8\mathrm{m/s}$ gust, before mimicking behaviour seen for the $14\mathrm{m/s}$

gust once pitch control becomes active. The local peak in above-rated axial loading, close to the point of maximum wind speed, is slightly larger for the $11\mathrm{m/s}$ gust. Rapid axial unloading again takes place, with radial force unloading occurring at around the same time. This indicates that for certain operating points a gust, or similar event, could lead to the complete (axial and radial) unloading and subsequent re-loading of a main-bearing row during operation. The maximum value of main-bearing row radial loading ($\sim 6000\mathrm{kN}$) from the three investigated gust events occurs during the $11\mathrm{m/s}$ gust. While additional interactions

are present as the wind speed varies during a gust, the general trend of radial loading vs. wind speed seen in steady operating results (Figure 4a) is still present in radial load gust results. Overall, non-steady gust interactions have been shown to drive significant load variations, with controller and structural dynamics, as well as response timescales, directly influencing the time-variations in main-bearing loads.



**Figure 12.** Radial and axial main-bearing loads on the upwind row of the overhung support configuration, along with control variable values, resulting from gusts at $v = 8$m/s (subfigures a-c), $v = 11$m/s (subfigures d and e) and $v = 14$m/s (subfigures f-h). Line segments in this figure correspond to those in Figure 11.





Figure 13 shows control variables and radial force responses to extreme wind direction change events of 30° in 6 seconds
when operating at 8m/s and 16m/s. Similar to when discussing yaw effects in Section 4.1, extreme direction changes from
a controls perspective are akin to a drop in wind speed. Responses shown in Figures 13b and 13c corroborate this, with the
pitch angle reducing during above-rated operation and torque reducing during below-rated operation. The general trends seen
in steady yaw cases are again visible here, with below-rated response driving changes principally in the horizontal plane and
above-rated principally in the vertical plane. With respect to the latter, resulting fluctuations in radial loading (in both magnitude
and direction) are significantly larger than would be expected based on steady yaw results alone (Figures 4c and 5). Interactions
between the wind conditions, control system and structural dynamics, therefore, appear to again be playing an important role
in determining the resulting characteristics of loading on the main-bearing. In both investigated cases of an extreme wind
direction change, the overall variability in radial loading (*i.e.* loop size) can be seen to increase as the event progresses.

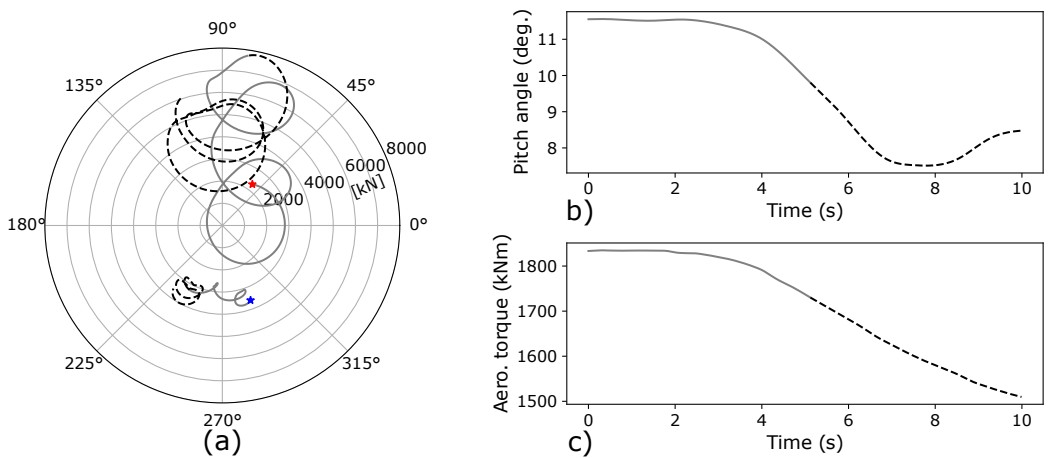

**Figure 13.** Radial main-bearing loads on the upwind row of the overhung support configuration, along with control variables values, resulting
from extreme wind direction change events. Subfigure (a) shows radial loads during operation at 8m/s (blue star) and 16m/s (red star).
Subfigures (b) and (c) show concurrent control variable values at 16m/s and 8m/s respectively.

## 5    Conclusions

This work has provided insight into the load characteristics of main-bearings in large, modern wind turbines. The applied
models are simple, so findings should be treated as being indicative as opposed to conclusive. A summary of key findings,
followed by further discussion on each, is:

1. Overhung support main-bearing radial loads were almost an order of magnitude larger than those for the centered support
main-bearing



2. In steady mean wind conditions, the reference height mean wind speed, shear exponent, yaw offset and veer gradient were all seen to drive changes in main-bearing mean loading and load variations. In particular, load variations were highly sensitive to the veer gradient

       3. In steady mean wind conditions, while control dynamics had been removed, the turbines' operational strategies (see Section 2.4) were seen to impact main-bearing load patterns

4. Analysis using upscaled turbine models indicated that main-bearing radial loads and load variations are increasing at least cubically with the turbine rotor radius

       5. During deterministic non-steady events, there were important interactions between wind conditions, dynamics/timescales of the wind turbine controller and the structural dynamics of the turbine itself that impacted main-bearing loads. Rapid main-bearing unloading and re-loading events were observed for gusts in above-rated conditions.

The observation in item 1 is evident from the model equations and parameters, but, is also intuitively true because overhung main-bearings are required to react the same rotor moments across a smaller distance. However, it does not necessarily follow that the centered support is superior, because each configuration will be designed to handle the loads it will experience. Furthermore, lifetime costs are what ultimately determines which might be optimal, with a number of factors contributing to this, including capital costs, compactness of drivetrain design, reliability, maintainability, and replaceability (Guo et al., 2017; Hart et al., 2020; Nejad et al., 2021). The observed differences in main-bearing load magnitudes may still be important when considering optimal configurations, but further work is needed before any conclusions relating to this might be reached. With respect to item 2, potentially the most important observation is the sensitivity of load fluctuations to veer, because veer is generally not considered during the drivetrain/turbine design process. Further investigation of the impacts of veer on main-bearing operation and lifetime is therefore recommended. Item 3 indicates that the design of a wind turbine's operational strategy may itself influence main-bearing reliability. For example, this might include where pitch control becomes active, whether this occurs as the turbine reaches rated power or before, and how switching between control regions is implemented. The upscaling findings outlined in item 4 present a potentially problematic picture of how main-bearing load characteristics may evolve as turbines become larger. It is again emphasised that the scaling employed in this paper accounts for size alone, and not changes in technology. Therefore, more work is needed to better understand how main-bearing loads will change for increasing sizes of real world turbines. The presented findings suggest that such investigations may have important implications for main-bearing design and reliability. From item 5, it follows that main-bearing load response must be considered in the context of full system dynamics, including those of the wind field, turbine and controller.

    The impacts of observed time-varying load behaviour on bearing internal stresses, deformations and lubrication should also be studied as part of future work. Such analysis will require more sophisticated modelling of these main-bearing configurations. Finally, it is also important to recognise the simplicity of wind field representations used in this study. The steady mean wind fields used in simulations should be interpreted as providing a mean-flow characterisation of an idealized atmospheric surface layer interacting with the modeled wind turbines during a typical afternoon with steady winds and relatively homoge-



neous surface conditions in the horizontal. Deterministic gusts and direction change events then introduced some non-steady behaviour into the flow which contain elements of turbulence eddy structure. While, as discussed in Section 2.3, these were

deemed sensible and reasonable simplifications for the purposes of this initial study, a rigorous characterisation of main-bearing loading in operating wind turbines will require the inclusion of true turbulent eddy structure. In particular, the non-steady event results presented here imply that main-bearing load response to the passage of turbulent eddies will be influenced by turbulence characteristic timescales relative to the timescales associated with the turbine's controller and structural dynamics. A field measurement campaign of loading, turbine operational variables and wind field structure, if possible, would allow similar

analyses to be performed for real world operation.

**Appendix A:  Formal definitions of modelled non-steady wind events**

**A1   Gusts**

Gusts are deterministic events where the wind speed rises rapidly to a peak before returning to the original wind speed. They are typically modelled as a linear superposition over the background wind conditions. The IEC 61400-1 wind turbine design

standard (IEC, 2005-08) defines Extreme Operating Gusts (EOGs) for standard wind turbine classes via the equation,

$$
V(z,t) = \begin{cases} V(z) - 0.37 V_{gust} \sin(3\pi t/T)(1 - \cos(2\pi t/T)), & \text{for } 0 \leq t \leq T \\ V(z), & \text{otherwise} \end{cases}
\tag{A1}
$$

where $V$ is the background wind speed, $z$ is height, $V_{gust}$ is the peak wind speed of the gust, $t$ is time and $T = 10.5\text{s}$. To understand and isolate the effect of gusts on main-bearing load response, simulations were conducted using a constant background wind field with a shear exponent of 0.2. Because wind speed varies during the gust event, it is necessary to keep

control active during these simulations.

**A2   Extreme wind direction change**

Extreme wind direction changes (EDC) are events in which the mean wind direction changes rapidly by tens of degrees. The characteristics of EDC events are specified in the IEC 61400-1 wind turbine design standard (IEC, 2005-08). First, the magnitude of the EDC, $\theta_e$ is defined via the relationship,

$$
\theta_e = \pm 4 \arctan\left( \frac{\sigma_1}{V_{hub}\left(1 + 0.1\left(\frac{D}{\Lambda_1}\right)\right)} \right)
\tag{A2}
$$

in which $\sigma_1$ is the standard deviation for the standard turbulence at the given mean wind speed ($V_{hub}$), $\Lambda_1$ is the turbulence scale parameter (42 for turbines above 60m hub height) and $D$ is the turbine rotor diameter. The transient is then given by,

$$
V(z,t) = \begin{cases} 0^o & \text{for } t < 0 \\ \pm 0.5\theta_e(1 - \cos(\pi t/T)) & \text{for } 0 \leq t \leq T \\ \theta_e, & \text{for } t > T \end{cases}
\tag{A3}
$$





where T = 6s is the time taken to complete the EDC. Over a six second period a wind turbine is unlikely to have begun yawing

to compensate for the direction change and, even if it began at the start of the event it would only have yawed a negligible 3

degrees at a typical yaw rate of 0.5 degrees per second. Hence, for the simulations performed here, it is assumed no yawing

occurs.

**Appendix B: Alternative derivation of force response equations**

Force response equations (Equations 1-3) used in the current work may be reached via an alternative derivation. While this

alternative route was deemed unnecessarily verbose for the main body of the manuscript, it provides helpful insights regarding

the applied equations and the quality of estimate they represent. A brief summary is therefore provided here, along with a

discussion of implications for the analysis presented in this paper.

The hub, driveshafts, and frame are again approximated as rigid. Following an approach similar to that of Stirling et al.

(2021), force and moment response behaviour at each bearing row (and in each plane) may be approximated using combined

linear and rotational springs of stiffness $K_r$ and $K_\theta$, respectively. Spring stiffness values are assumed equal in horizontal and

vertical planes and at each bearing row. The hub itself is constrained to two degrees of freedom in each plane, these being

radial displacement ($\delta_r$) of the hub centreline and rotation ($\delta_\theta$) about the midpoint between bearing rows. It is further assumed

that hub rotational displacements are small. With the above in place it is straightforward to express the main-bearing row radial

($\delta_{r_i}$) and rotational ($\delta_{\theta_i} = \delta_\theta$) deflections, in each plane, as functions of the applied loads, stiffness values and parameters $L_h$

and $L_b$. Main-bearing radial force and moment response equations are then easily obtained and have the following form,

$$F_i^* = -\delta_{r_i}^* K_r = -\frac{1}{2}\left(\left[1 \pm L_h\left(\frac{K_r L_b}{K_\theta + K_r L_b^2}\right)\right]F_h^* \pm \left(\frac{K_r L_b}{K_\theta + K_r L_b^2}\right)M_h^*\right) \tag{B1}$$

$$M_i^* = -\delta_\theta^* K_\theta = \pm\frac{1}{2}\left(\frac{K_\theta}{K_\theta + K_r L_b^2}\right)M_h^*, \tag{B2}$$

for $i = 1, 2$ and where each $*$ symbol is a placeholder for an axis label, $y$ or $z$. If appropriate stiffness values, $K_r$ and $K_\theta$ are

known, the above equations may be resolved to estimate force and moment responses at each row. In the current work sensible

stiffness values were not known, making further simplification necessary. From the rolling bearing literature (Tong and Hong,

2014; Zhang et al., 2019), values of $K_r L_b^2$ will generally be a factor of $10^3$ larger than $K_\theta$ if $L_b$ is of order 1m. Therefore, in

general, $K_\theta/K_r L_b^2 \ll 1$. It follows that without further information the approximation,

$$K_\theta + K_r L_b^2 \approx K_r L_b^2, \tag{B3}$$

is not unreasonable. Similarly, if $L_b$ is of order $10^{-1}$m then $K_r L_b^2$ is expected to be a factor of 10 larger than $K_\theta$. The same

approximation remains viable, albeit of poorer quality, in this latter case. Note these two cases correspond to the modelled

centered and overhung support systems of the current work, respectively. Applying the approximation (Equation B3) to load

response estimates (Equations B1 and B2), the following may be observed:

1. $K_r L_b$ terms in radial force response expressions cancel, removing the dependence on stiffness values and recovering the

   radial force equations of Section 3.3 (Equations 2 and 3)





2. The moment response estimate becomes $M_i^* = \pm\frac{1}{2}\left(\frac{K_\theta}{K_r L_b^2}\right)M_h^*$ which for large $M_h^*$ will be non-negligible even when $K_\theta/K_r L_b^2$ is small.

In the context of the current paper, this short analysis provides a more quantitative assessment of model accuracy, with the necessary approximations being of poorer quality (while remaining viable) for the overhung support case. In addition, while force response is expected to be the main support mechanism, it has also been shown that large enough hub moments will

still elicit a non-negligible moment response across individual bearing rows. Importantly, the above analysis demonstrates that the force response equations (Equations 2 and 3) used in this work remain valid even in such cases. For the overhung support configuration in particular, potentially damaging differential loading across moment supporting rollers appears a distinct possibility. Finally, it is emphasised that while valuable and informative, the main-bearing representation outlined here remains a significant simplification in which much is not accounted for. For example, in reality differences in stiffness values between

main-bearing rows would be expected due to one row reacting thrust. Findings must therefore be interpreted in the context of these being simplified engineering representations of these systems.

## Appendix C: Further upscaling results

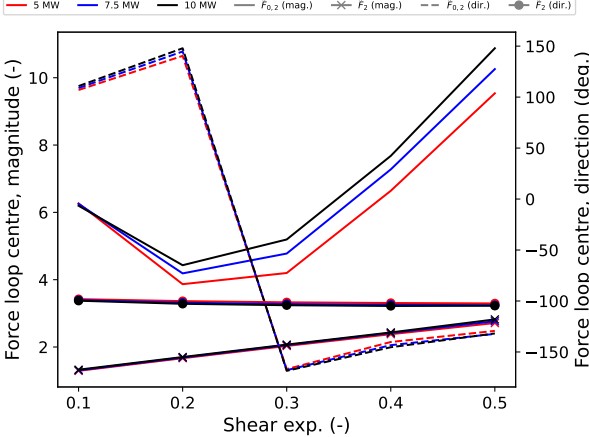

**Figure C1.** Non-dimensionalised, downwind row radial force-loop-centre magnitude and direction values for overhung and centered main-bearing configurations across the three modelled turbines as the shear exponent is varied.

*Competing interests.* The authors declare they have no competing interests.

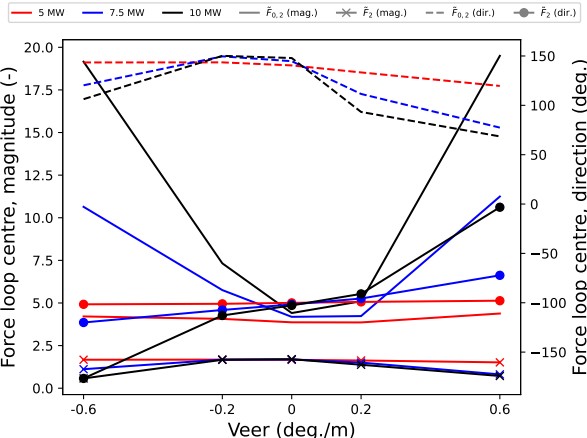

**Figure C2.** Non-dimensionalised, downwind row radial force-loop-centre magnitude and direction values for overhung and centered main-bearing configurations across the three modelled turbines as the veer gradient is varied.

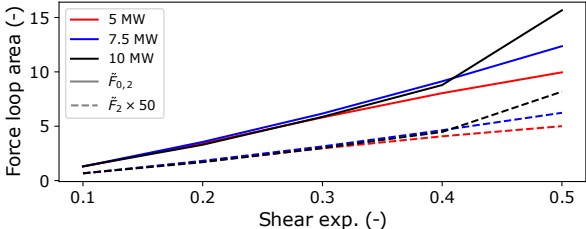

**Figure C3.** Non-dimensionalised downwind row radial load-loop-area results for various shear exponents for overhung and centered support main-bearing configurations across the three modelled turbines.

*Acknowledgements.* E. Hart is funded by a Brunel Fellowship from the Royal Commission for the Exhibition of 1851. This work was
also authored in part by the National Renewable Energy Laboratory operated by the Alliance for Sustainable Energy, LLC, for the US
Department of Energy (DOE) under contract no. DE-AC36-08GO28308. Funding was provided by the US Department of Energy Office of
Energy Efficiency and Renewable Energy Wind Energy Technologies.

WIND ENERGY SCIENCE DISCUSSIONS
european academy of wind energy
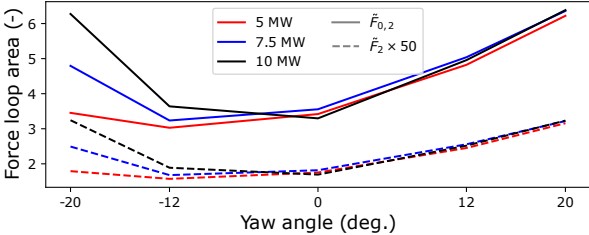

**Figure C4.** Non-dimensionalised downwind row radial load-loop-area results for various yaw angles for overhung and centered support main-bearing configurations across the three modelled turbines.

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
