# Peer review of "Impacts of wind field characteristics and non-steady deterministic wind events on time-varying main-bearing loads"

_Wind Energy Science, 2022_

## Author Comment (AC1)

**wes-2022-1**: Impacts of wind field characteristics and non-steady deterministic wind events on time varying main-bearing loads

**Response to Reviewer 1**

Dear Reviewer,

Thank you for taking the time to consider our paper, and for making valuable suggestions regarding how it might be improved. We include your comments below in **blue**, followed by our responses in **black**.

Thank you for the nice publication, there are some minor spelling issues.

We will carefully review spelling and grammar throughout the paper prior to submitting the revised version.

Additionally, there some minor flaws.

- all figures are missing light grids

  We will look to add grids to figures where it will enhance interpretability and readability. Note, in some cases (e.g. dimensionless upscaling plots) we are concerned a grid might make the plots appear to 'busy'. We will check, and where that is felt to be the case we will keep the current form.

- p.1 l.11: wind turbines with 3MW offshore are past the state of art. Such a small size is currently erected onshore

  Here we are simply quoting the range as described in the cited literature (Barter et al., 2020). In addition, much of the existing main-bearing work has, to date, focussed on machines of power ratings less than 3MW, hence in that particular context there is still plenty of scope for improved understanding and scientific contributions from 3MW and upwards. We do take you point here, but, we feel it is appropriate to leave the stated range as it is, for the reasons given here.

- p.4 l.100: the paper would be more comprehensible, if some wind turbine characteristics would be stated (ex. rotor diameter, rated torque, rated wind speed)

  This is an excellent point. We will add a brief summary of such details when revising the manuscript.

- p.5 l.124: loads are modeled according to IEC 61400-1 but is only mentioned far later or in the appendix. BTW. why not use the current IEC 61400?

  We will reassess when we mention the standards as it may be relevant to point to them sooner, as you suggest. Thank you for pointing out that we are currently citing the old version of the standards. We do use the current standard, this is simply an error in the reference info. We will update the reference accordingly.

- p.6 figure 1: Thrust curve consists out of 3 segments before rated conditions, explanation missing

The three sections in the thrust curve relate to the two constant speed regions and a variable speed 'max aero-efficiency' tracking region. This is all quite standard for wind turbines, and so we hadn't gone into detail regrading this breakdown in the current paper. However, perhaps some more info would be useful and so we will consider how to provide more information when revising the paper. In particular, we will ensure the reader is pointed towards relevant literature which provides a more detailed description.

- p.8 l.201: a rigid shaft is quite a simplification. The reasonable justification is missing

This relates to the overall aims of the paper, the information available for modelling and the context in which readers are encouraged to interpret presented results. Crucially, this paper explicitly states that it is looking to explore "… characteristics, relative magnitudes and drivers" (Line 83) of main-bearing loads in these machines. Therefore, it is not claimed that the presented model exactly reproduces load responses at the main-bearing, but that it allows us to estimate these loads and load behaviours well enough to gain insights into the operating conditions of these components. Furthermore, we do justify this claim by indicating recent work where similar model have been shown to provide reasonable estimates of loading (including using those which assume rigidity of the shaft) "Models of this type have previously been shown to be able to reproduce main-bearing load reactions calculated using higher fidelity finite element representations of a shaft/main-bearing system (Stirling et al., 2021). Depending on model specifics, mean percentage errors of between 1.54% and 22.74% were reported." (Line 236). In addition, shaft-drivetrain systems in these machines are often required to prevent excessive displacements due to system tolerances, in particular in direct drive cases where the main-bearing is required to maintain generator air-gaps within their operating limits. Hence, these systems are designed to be very 'stiff' in the first place. But, again we are careful to caveat our results for the reader, e.g. "The presented models should therefore be interpreted as providing first order engineering estimates of load response for the main-bearings in question. Inline with the stated aims of this work, such models allow for the characteristics, drivers, and general orders of magnitude of main-bearing loading to be investigated for these machines. As such, they are sufficient for the work undertaken in the current paper. Model limitations should, however, be kept in mind when interpreting results." (Line 239). We therefore feel that the context and limitations of the current models are made clear to the reader, but, also that (for the reasons outlined here and in the paper) the presented model is sufficient for the aims and scope of this paper.

- p.10 l.265: you mention that you are using a reference load. Would that not mean that the output loads in the figure 4 are non-dimensional

No. The 'reference load' is provided to give the reader an idea of the nominal loading which might be expected to occur for that component. This assists with providing context for results. But, we felt the actual load values were important to include (in kN) on these particular plots, since then readers get a sense for what the real load inputs going into these bearings are.

- p.11 figure: 4: axis-labels are missing; please indicate the direction of the rotor weight, as 0° points towards a right.

Axis labels aren't missing. Both the radial magnitude units (kN) and directional units (degrees) are indicated in the plots. The reference frame being used, and hence the direction of rotor weight, is described in the opening paragraph of Section 4. When revising the manuscript we will consider whether an additional sentence or two might help connect these things together here for the sake of clarity.

- p.15 figure 7: Not traceable. Labels cannot be read. Maybe split the graphic into two. It is not clear which is centered, and which is overhang

The notation for centred and overhung loads is described in the opening paragraph of Section 4. We will try and enlarge the legend labels when revising the paper. Regarding traceability, we will see if these figures are improved by the addition of a grid, although as stated previously we will check to ensure this doesn't make things too 'busy'.

- p.15 l.348: explanation of how loop area is determined would be nice. As for my understanding: It is the area enclosed as shown in figure 4

This is indicated by a footnote in Line 348, but we will look to expand this a little to explain more clearly. Your interpretation is about right. Individual loops are identified in load time-series before an ellipse is fitted to them. The area of the loop is taken to be the area of this best-fitting ellipse. This is all based on previous work cited in the paper.

- p.19 figure:12: ideal thrust curve should be indicated differently, otherwise it is like the end of the gust

Excellent point. This is just an error in the plotting. The start of the time-varying thrust curve should be grey (rather than black) to match the other plots in Fig 12. We will correct this.

- p.20 l.405: perspective of centered support missing?

As shown earlier in the paper the behaviour of the two supports is very similar, with the order of magnitude the only major difference really. Therefore, centred support results may be inferred from those presented here for the single support.

- p.20 figure: 13: You show the control variable for corresponding wind speed. Please ad the corresponding star in control variable to improve clarity

Excellent point. We will add this in as suggested.

- p.21 l.420 cubical increase, is not clear as rotor diameter missing

The logic behind the observed cubic increase in loads is outlined in Section 4.2 "… Force results for all turbines were therefore non-dimensionalised using half of their respective rotor weights in each case. In order to consider load variability, ellipses were fitted to the identified load loops, with elliptical areas then calculated (Hart, 2020). Load loop area results were non-dimensionalised using the square of the W/2 value of each turbine. Recall that on upscaled models the rotor weights scale cubically with the turbine rotor radius (see Section 2.2); therefore, overlapping lines in dimensionless results plots for the three models indicate that main-bearing loads are also increasing cubically with the rotor radius." (Line 328). Hence, one does not need to explicitly know the rotor Radii in order for this conclusion to be reached. We will be

adding rotor Radii values in response to a previous comment, but we feel it is important to highlight why we are able to reach this particular conclusion and where that discussion is presented in the paper. We will look to link back to that discussion in the conclusions to ensure clarity.

- p.21 conclusion: Your conclusion contains discussions and new references. A conclusion should only base on the work shown earlier. A split between discussion and conclusion might be better

The current format of the Conclusions section was selected as it seemed to provide the best balance in terms of insight and clarity for the reader, while also avoiding repetition. We will consider whether a split as you suggest might be helpful, or whether to stick with the current format.

---

## Author Comment (AC2)

**wes-2022-1**: Impacts of wind field characteristics and non-steady deterministic wind events on time varying main-bearing loads

**Response to Reviewer 2**

Dear Reviewer,

Thank you for taking the time to consider our paper, and for making valuable suggestions regarding how it might be improved. We include your comments below in **blue**, followed by our responses in **black**.

The publication is interesting, where looking at reduced order models for main bearing load estimation is something that could be used for various applications.

However, it would require a bit more elaboration on the turbine characteristics used, like mentioned by the previous referee.

Yes, we agree this would be useful for readers. We will provide a summary of turbine model information for the 5, 7.5 and 10 MW turbines.

P1. l.21 seems to have a typo, or at least reads a bit weird.

Yes, you are correct that this is a typo. It should read "in which the entire hub rotates about a stationary internal mounting with one or more main-bearings". We will correct this.

P7. figure 2: figure 3 explains perfectly how you calculate the bearing loads, however from figure 2 it might be hard for readers to understand the visual difference. You refer to GE, which shows a very in depth figure, but maybe consider changing figure 2 to make it visually more understandable for readers who have not seen these configurations before (up to the author). I would also just make a 2nd figure of the centered support in figure 3 if I were you.

We will consider if any improvements can be made here regarding representation of the drivetrains. However, please note that in a long paper which already has a large number of figures we are trying to avoid adding more unless strictly necessary. In addition, please note that Fig 3 applies equally to both configurations depending on whether Lh>Lb (overhung support) or Lh = 0 (centred support). Therefore, both are included here implicitly. We will add a note to the caption of Fig. 3 to make sure this point is clear.

P8. l. 220 Regarding the Equations , it is nice to mention that you assume static equilibrium, rather than dynamic equilibrium.

P15. figure 7,8: The legend size should be larger.

We will try and improve readability of the legends in these figures.

Your results section reads like a results & discussion. It seems this was intentional, but it is not mentioned. Either split these two up or change the title of the results section to results and discussion.

In our experience a results section will generally contain discussion as well. But, we will consider whether an altered section title might be helpful here.

The use of grids in figures can be nice. The paper and some sentences seem to be (perhaps excessively) long and could be shortened. Text becomes more understandable for readers when sentences are kept short. However, this should be up to the author.

These are all excellent points. We will consider whether grids in some figures can help the reader, while also avoiding those figures becoming too 'busy'. We will also review the paper in general, shortening sentences if possible.

---

## Author Response (AR1)

**wes-2022-1**: Impacts of wind field characteristics and non-steady deterministic wind events on time varying main-bearing loads

**Author response**

Contents:

(A) Response to Editor

(B) Response to Reviewer 1

(C) Response to Reviewer 2

**wes-2022-1**: Impacts of wind field characteristics and non-steady deterministic wind events on time varying main-bearing loads

**Response to Editor (post revisions)**

Dear Amir,

Thank you for handling the submission of this paper to WES.

Both reviewers made excellent suggestions for how we might improve the manuscript and ensure clarity for readers. We have acted on their suggestions and revised the manuscript accordingly. Detailed and updated responses to both reviewers are included below.

Best regards,

Edward Hart (on behalf of all co-authors)

**wes-2022-1**: Impacts of wind field characteristics and non-steady deterministic wind events on time varying main-bearing loads

**Updated Response to Reviewer 1 (post revisions)**

Dear Reviewer,

Thank you for taking the time to consider our paper, and for making valuable suggestions regarding how it might be improved. We include your comments below in **blue**, followed by our updated responses in **black** (now the suggested edits have been made).

Thank you for the nice publication, there are some minor spelling issues.

Thank you. We have been through the document carefully to correct all spelling issues.

Additionally, there some minor flaws.

- all figures are missing light grids

    Light grids have now been added to all figures where we felt it was appropriate (e.g. a grid doesn't work for figures in which there are two y-axes, such as in our non-dimensional plots).

- p.1 l.11: wind turbines with 3MW offshore are past the state of art. Such a small size is currently erected onshore

    Here we are simply quoting the range as described in the cited literature (Barter et al., 2020). In addition, much of the existing main-bearing work has, to date, focussed on machines of power ratings less than 3MW, hence in that particular context there is still plenty of scope for improved understanding and scientific contributions from 3MW and upwards. We do take you point here, but, we feel it is appropriate to leave the stated range as it is, for the reasons given here.

- p.4 l.100: the paper would be more comprehensible, if some wind turbine characteristics would be stated (ex. rotor diameter, rated torque, rated wind speed)

    This is an excellent point. We have therefore added a table (Table 1, Page 4 in the revised manuscript) providing summary information for the three turbine models.

- p.5 l.124: loads are modeled according to IEC 61400-1 but is only mentioned far later or in the appendix. BTW. why not use the current IEC 61400?

    Thank you for pointing this out, we have added an earlier reference to the standards to make this clear sooner in the paper (see Page 5, Line 129 of the revised manuscript). Thank you for pointing out that we were citing the old version of the standards. We do use the current standard, this was simply an error in the reference info. We have updated the reference accordingly.

- p.6 figure 1: Thrust curve consists out of 3 segments before rated conditions, explanation missing

The following sentence has been added to indicate the fact that there are multiple regions in the operating strategy, with the reader pointed to resources containing further information. The added sentence reads "Such operational strategies contain a number of distinct regions, for a more detailed discussion see Hart et al. (2020); Jenkins et al. (2021)" Page 6, Line 146.

- p.8 l.201: a rigid shaft is quite a simplification. The reasonable justification is missing

This relates to the overall aims of the paper, the information available for modelling and the context in which readers are encouraged to interpret presented results. Crucially, this paper explicitly states that it is looking to explore "… characteristics, relative magnitudes and drivers" of main-bearing loads in these machines. Therefore, it is not claimed that the presented model exactly reproduces load responses at the main-bearing, but that it allows us to estimate these loads and load behaviours well enough to gain insights into the operating conditions of these components. Furthermore, we do justify this claim by indicating recent work where similar model have been shown to provide reasonable estimates of loading (including using those which assume rigidity of the shaft) "Models of this type have previously been shown to be able to reproduce main-bearing load reactions calculated using higher fidelity finite element representations of a shaft/main-bearing system (Stirling et al., 2021). Depending on model specifics, mean percentage errors of between 1.54% and 22.74% were reported.". In addition, shaft-drivetrain systems in these machines are often required to prevent excessive displacements due to system tolerances, in particular in direct drive cases where the main-bearing is required to maintain generator air-gaps within their operating limits. Hence, these systems are designed to be very 'stiff' in the first place. But, again we are careful to caveat our results for the reader, e.g. "The presented models should therefore be interpreted as providing first order engineering estimates of load response for the main-bearings in question. Inline with the stated aims of this work, such models allow for the characteristics, drivers, and general orders of magnitude of main-bearing loading to be investigated for these machines. As such, they are sufficient for the work undertaken in the current paper. Model limitations should, however, be kept in mind when interpreting results.". We therefore feel that the context and limitations of the current models are made clear to the reader, but, also that (for the reasons outlined here and in the paper) the presented model is sufficient for the aims and scope of this paper.

- p.10 l.265: you mention that you are using a reference load. Would that not mean that the output loads in the figure 4 are non-dimensional

No. The 'reference load' is provided to give the reader an idea of the nominal loading which might be expected to occur for that component. This assists with providing context for results. But, we felt the actual load values were important to include (in kN) on these particular plots, since then readers get a sense for what the real load inputs going into these bearings are.

- p.11 figure: 4: axis-labels are missing; please indicate the direction of the rotor weight, as 0° points towards a right.

Axis labels aren't missing. Both the radial magnitude units (kN) and directional units (degrees) are indicated in the plots. The reference frame being used, and hence the direction of rotor weight, is described in the opening paragraph of Section 4. However, we agree that it is useful to signpost this again for the reader when the plots are

shown. Therefore, the following sentence has been added to the caption of Figures 4 and 5: "As described in the opening paragraph of Section 4, force results are aligned with the true direction of applied force when viewing the turbine from upwind. For example, an applied force acting vertically downwards corresponds to an angle of 270° in the above plots."

- p.15 figure 7: Not traceable. Labels cannot be read. Maybe split the graphic into two. It is not clear which is centered, and which is overhang

We have enlarged the legend on dimensionless force plots to ensure labels can be clearly read. We have also added a note in the caption to remind the reader which corresponds to the overhung support and which corresponds to the centered support. We believe that it is better to keep these figures as single figures, rather than splitting each into two, as the key information is made accessible without massively increasing the total number of figures in the paper. Since there are two axes a grid cannot be added, but, we feel that the key findings from these results are easily taken from these figures without a grid present.

- p.15 l.348: explanation of how loop area is determined would be nice. As for my understanding: It is the area enclosed as shown in figure 4

This is described (including relevant literature reference) in the opening paragraph of Section 4.2. We have therefore indicated this in the footnote at the start of Section 4.2.2, Page 15, Line 353.

- p.19 figure:12: ideal thrust curve should be indicated differently, otherwise it is like the end of the gust

Excellent point. You are correct there was an issue in this plot. To ensure any confusion is avoided we have changed the ideal thrust curve to be magenta. We have also made this change in Fig 1 for the sake of consistency.

- p.20 l.405: perspective of centered support missing?

As shown earlier in the paper the behaviour of the two supports is very similar, with the order of magnitude the only major difference really. Therefore, centred support results may be inferred from those presented here for the single support.

- p.20 figure: 13: You show the control variable for corresponding wind speed. Please ad the corresponding star in control variable to improve clarity

Excellent point. We have done this as suggested. In addition, we added stars at the beginning of control variable plots in Figure 12 to ensure consistency throughout the paper. To avoid any confusion between the cases in Figs 12 and Figs 13 we have also made sure the stars in each are different colours.

- p.21 l.420 cubical increase, is not clear as rotor diameter missing

The logic behind the observed cubic increase in loads is outlined in Section 4.2 "… Force results for all turbines were therefore non-dimensionalised using half of their respective rotor weights in each case. In order to consider load variability, ellipses were fitted to the identified load loops, with elliptical areas then calculated (Hart,

2020). Load loop area results were non-dimensionalised using the square of the W/2 value of each turbine. Recall that on upscaled models the rotor weights scale cubically with the turbine rotor radius (see Section 2.2); therefore, overlapping lines in dimensionless results plots for the three models indicate that main-bearing loads are also increasing cubically with the rotor radius.". Hence, one does not need to explicitly know the rotor radii in order for this conclusion to be reached. We have added rotor radii values as suggested in a different comment, but we feel it is important to highlight why we are able to reach this particular conclusion and where that discussion is presented in the paper. To try and ensure clarity for the reader, we have linked back to the relevant logic on cubic scaling when describing results in Sections 4.2.1 and 4.2.2.

- p.21 conclusion: Your conclusion contains discussions and new references. A conclusion should only base on the work shown earlier. A split between discussion and conclusion might be better

On reflection, we have decided to stick with the current form of conclusion, as we feel this clearly encapsulates the main findings and recommendations of the paper.

**wes-2022-1**: Impacts of wind field characteristics and non-steady deterministic wind events on time varying main-bearing loads

**Updated Response to Reviewer 2 (post revisions)**

Dear Reviewer,

Thank you for taking the time to consider our paper, and for making valuable suggestions regarding how it might be improved. We include your comments below in **blue**, followed by our updated responses in **black** (now the suggested edits have been made).

The publication is interesting, where looking at reduced order models for main bearing load estimation is something that could be used for various applications.

However, it would require a bit more elaboration on the turbine characteristics used, like mentioned by the previous referee.

Yes, we agree this would be useful for readers. We have therefore added a table (Table 1, Page 4 in the revised manuscript) providing summary information for the three turbine models.

P1. l.21 seems to have a typo, or at least reads a bit weird.

Yes, you are correct that this is a typo. It should read "in which the entire hub rotates about a stationary internal mounting with one or more main-bearings". We have corrected this.  Page 1, Line 21

P7. figure 2: figure 3 explains perfectly how you calculate the bearing loads, however from figure 2 it might be hard for readers to understand the visual difference. You refer to GE, which shows a very in depth figure, but maybe consider changing figure 2 to make it visually more understandable for readers who have not seen these configurations before (up to the author).  I would also just make a 2nd figure of the centered support in figure 3 if I were you.

We understand your point, however, the diagrams in Fig. 2 were kept deliberately simplistic since these models are relatively general and may represent a range of different technologies. As such, we prefer to keep the representation similarly general in these diagrams. As you point out, detailed depictions of the various possibilities are available in the cited literature (both in the GE case and for the others as well).

Please note that Fig 3 does already apply equally to both configurations depending on whether $L_h > L_b$ (overhung support) or $L_h = 0$ (centred support). Therefore, both are included here implicitly. We have added a sentence to the manuscript which makes this point clear: "An overhung support corresponds to the case $L_h > L_b$, while a centered support is obtained when $L_h = 0$." Page 8, Line 199

P8. l. 220 Regarding the Equations , it is nice to mention that you assume static equilibrium, rather than dynamic equilibrium.

We agree, and so have added this into the relevant sentence: "With the above approximations in place, static equilibrium force responses at each bearing row for the generic drivetrain shown in Figure 3 are given by…" Page 9, Line 220

P15. figure 7,8: The legend size should be larger.

All legends have been enlarged in figures 7, 8, C1 and C2.

Your results section reads like a results & discussion. It seems this was intentional, but it is not mentioned. Either split these two up or change the title of the results section to results and discussion.

We have changed the title of the section to 'Results and discussion' as suggested.

The use of grids in figures can be nice. The paper and some sentences seem to be (perhaps excessively) long and could be shortened. Text becomes more understandable for readers when sentences are kept short. However, this should be up to the author.

These are all excellent points. Light grids have now been added to all figures where we felt it was appropriate (e.g. a grid doesn't work for figures in which there are two y-axes, such as in our non-dimensional plots). We have also reviewed the paper generally to try and ensure clarity.